# Effects of Visual Communication Design Accessibility (VCDA) Guidelines for Low Vision on Public and Open Government Health Data

**DOI:** 10.3390/healthcare11071047

**Published:** 2023-04-06

**Authors:** Jongho Lim, Woojin Kim, Ilkon Kim, Eunjoo Lee

**Affiliations:** School of Computer Science & Engineering, College of IT Engineering, Kyungpook National University, Daegu 41566, Republic of Korea

**Keywords:** open government data, healthcare, low vision

## Abstract

Since 2019, the Korean government’s investments in making data more accessible to the public have grown by 337%. However, open government data, which should be accessible to everyone, are not entirely accessible to people with low vision, who represent an information-vulnerable class. Emergencies, such as the COVID-19 pandemic, decrease face-to-face encounters and inevitably increase untact encounters. Thus, the information gap experienced by low-vision people, who are underprivileged in terms of information, will be further widened, and they may consequently face various disadvantages. This study proposed visual communication design accessibility (VCDA) guidelines for people with low vision. Introduced screens enhanced by accessibility guidelines were presented to 16 people with low vision and 16 people with normal vision and the speed of visual information recognition was analyzed. No statistically significant difference (*p* > 0.05) was found due to the small sample size; however, this study’s results approached significance with improved visual recognition speed for people with low vision after adopting VCDA. As a result of the intervention, the visual information recognition speed of both normal and low-vision people improved. Thus, our results can help improve information recognition speed among people with normal and low vision.

## 1. Introduction

According to the World Health Organization (WHO), 2.2 billion people worldwide are visually impaired, of whom 1.2 billion have low vision [1]. Despite there being numerous people with low vision, there are no information accessibility guidelines for this population [2].

In the past, the possession of information was the criterion for judging individual abilities, but in modern society, the ability to judge the quality of information has become a necessity [3,4]. When an epidemic such as COVID-19 occurs, “face-to-face” encounters decrease and “untact” encounters inevitably increase [5,6]. In this situation, the information gap of low-vision people, who are disadvantaged regarding access to information, will be further widened, and as a result, they may be exposed to various disadvantages [7,8].

Open government data refers to data or information processed optically or electronically that is generated or acquired by a public institution for purposes prescribed by laws and regulations, such as for databases and electronic files [9,10,11].

Korea has been one of the leading countries in the world for sharing open government data while maintaining quarantine policy during the COVID-19 pandemic [12]. In the “OECD Open-Useful-Reusable Government Data (OURdata) Index,” which provides information on the open government data policies and practices of member countries and partners of the Organization for Economic Cooperation and Development (OECD), Korea was ranked first in 2015, 2017, and 2019 [13,14]. As open government data allow people to actively participate in policies, the Korean government recognizes its value and enacts laws every year to improve the volume and quality of open government data [15]. Compared to 2019, the amount of open government data to be disclosed by public institutions has increased by approximately 337% [16].

However, open government data, which should be accessible to everyone, are not entirely accessible to people with low vision, who represent an information-vulnerable class. The lack of accessibility tools for people with low vision is widening the information divide [17,18].

Until now, studies on people with low vision have been lacking. There have been several studies on visually impaired people. For instance, Madiah et al. [19] suggested that text size, color and background could be influential factors for the recognition of text information for 10 children studied with partial vision. However, they did not examine the specific effects of text size, color, and background.

Additionally, Kim et al. [20] developed guidelines to improve user experiences on web pages and tested their effects in terms of the speed and correctness of recognition of information presented on a web screen in a sample of 26 visually impaired adults. However, Kim et al.’s study focused only on user experience (UX), and not on the graphic user interface (GUI). Text-oriented government data, mainly text information, has not been well supported, and limits the information accessibility of open government data for people with low vision.

More recently, Morento et al. [21] examined the effect of use of a screen magnification tool amongst 12 low-vision adults using an experimental design. They found that the speed of information recognition differed according to individual preference and did not identify what factors could improve information recognition speed for low-vision adults.

None of these studies examined the effect of the GUI, nor did they suggest ways to improve both text and visual information recognition speed. To solve this problem, this study developed GUI visual information accessibility guidelines for people with low vision. The GUI visual information accessibility guidelines for people with low vision aim to improve their ability to acquire information at the same level as those with normal vision.

This study was conducted according to the following procedure to present GUI guidelines for improving accessibility to public data for the low-visibility class: (1) Development of public data accessibility guidelines for people with low vision; (2) Development of test apps to validate the proposed guidelines; (3) Verification through experiment in a sample of low vision (16 people) and normal vision people (16 people).

## 2. Materials and Methods

### 2.1. Visual Communication Design Accessibility (VCDA) Guidelines for Low Vision

In the early 2000s, the World Wide Web Consortium’s Web 2.0 started with the goal of providing information equally to everyone [22]. Now, the Web 3.0 era allows users to take ownership of their data [23]. The Consortium established Web Content Accessibility Guidelines (WCAG) and Mobile Accessibility Guidelines (MWAG) for web and mobile accessibility [24,25].

However, MWAG and WCAG consider accessibility for anyone with a “physical” disability. Much of it feels very abstract from the perspective of developers and end users. Depending on the type and severity of the disability, it is often not helpful [26,27].

To solve this, user experience design guidelines (UXDG) were developed for the visually impaired, based on WCAG and MWAG in related studies [28]. However, the UXDG do not cater to the needs of users with low vision and are unsuitable for application to government public data, which is text-oriented (Table 1).

Taking the UXDG further and making them more inclusive, this study proposes visual communication design accessibility (VCDA) guidelines for low vision to improve the accessibility of open government data for people with low vision (Table 2).

The VCDA guidelines for people with low vision, shown in Table 2, propose focusing on text because most of the visual information provided by public data apps in Korea are provided in text. Additionally, VCDA was developed for people with low vision and specifies details that are not provided by MWAG and UXDG. For example, MWAG and UXDG are provided with simple statements that direct options, such as resizing or making text bold, to be considered.

In contrast, the VCDA guidelines proposed in this study specify that font size should be provided in three stages. This enables planners and developers to provide a consistent layout design that satisfies accessibility guidelines for people with low vision and provides aesthetic satisfaction to people with normal vision by clearly setting the layout design when developing an app.

The biggest advantage of a consistent layout is the ability to utilize the TTS function provided by the operating system of the mobile device. The basic principle of existing TTS is that text is converted into voice sequentially from the upper left to the lower right, just as a person reads the text with their eyes. If a consistent layout cannot be provided, and if TTS is used for information output in different layouts each time the screen of the app is switched, it is likely that completely different information will be provided. To prevent such problems, all items in the VCDA guidelines for people with low vision are written in a manner that maintains a consistent layout.

The VCDA guidelines proposed in this study were prepared by consulting experts who had researched visual impairment for 10 years and seeking additional advice from people with visual impairments. The details of the VCDA guidelines proposed in this study are as follows:(a)Text size and thickness support more than three options:

As of 2022, this has been provided by over-the-top (OTT) services (e.g., Netflix, Amazon Prime, etc.) [28]. Although these OTT services have different display sizes and resolutions when using PC web/apps and mobile web/apps, to provide the same text information, the size and thickness of characters are divided into three levels. For example, if the font size is provided to the user in an analog format (freely transformed), the layout will inevitably be transformed as the user adjusts the font size. Thus, there is a strong possibility that it cannot provide a clear screen.
(b)Designated background colors contrasted with white characters:

This is when the text is white, the background is black, and the readability of people with poor vision can be improved by arranging basic complementary colors (e.g., red-green, yellow-blue). This option must be supported because there are people who can only see it in this way (patients with specific visual diseases, e.g., patients with retinitis pigmentosa) [29].
(c)Batch layout considering TTS:

All TTS functions start reading from the top left to the bottom right, in column and row order. Therefore, changing the design of each page’s public data app will result in the information being read out of order. Therefore, the design must be configured under the premise that all pages can use TTS because of reduced accessibility to key information.
(d)Output alternate text or provide a TTS-listening button:

All media content must have an option to output textual information about what the content contains, considering accessibility.
(e)The characters in the image are recognized by optical character recognition (OCR) systems:

OCR is the process of converting analog information scanned through an optical lens into digital information. In other words, it is necessary to regulate the use of fonts so that machines can easily recognize them (e.g., Sans Serif (Gothic series) fonts have the highest OCR recognition rate. The handwriting-type serif series has a low recognition rate).
(f)Provide a text-only button if the body text of the information, such as media, is supported:

When text is supported, such as in images and videos (e.g., blog format), a reading mode for people with low vision must be provided so that users can access only the text. In the case of reading mode, specific menus are hidden, or the content of the text is highlighted, so it is advantageous not only for reading with residual eyesight but also for using the TTS function.
(g)If a video is included, notify the user about its playability as alternative text:

When the actual playable button and the advertising play button image are mixed, users with low vision do not know what to click. Even in public data apps, if an executable video is displayed as a simple icon, users may not know if they need to click it. If it is marked as a separate text in such a case, it can be accessed using TTS.
(h)State the current page position in a consistent position of the entire information:

Display the “current location” at a specific location desired by the user, such as the upper-left corner or the lower center.
(i)Always provide the same menu (e.g., Home or Top) to return to the first place.

Based on the three-step rule, a button to return to the beginning is provided so that people with low vision can re-enter from the beginning.
(j)The input area of all buttons is defined as 1.5 times the size of the button:

By increasing the size by 1.5 times, the interval between each button should be widened naturally, thereby helping prevent erroneous input by a person with low vision.
(k)Provide a text-oriented black-and-white screen based on the user experience:

On a black-and-white screen, all texts are distinguished only by contrast, not by color. The advantage of this is TTS; however, when using functions such as color inversion/smart inversion supported by mobile devices, it becomes possible to acquire information through conversion into a form desired by the user. For example, smart inversion is a function that reverses text, except for the image. If the text has color, activating the inversion function may cause side effects, such as invisible text. At this time, if the original source is classified only in black and white, text information can be recognized easily because there is no inversion effect due to the color.

### 2.2. Experimental Design

#### 2.2.1. Participants

In this study, 16 people with low vision were recruited from an online low-vision community, and 16 people with normal vision were recruited through social media to verify the developed guidelines.

Sample sizes were calculated based on an effect size of 0.6, power of 0.7, and probability of alpha error of 0.05 for a paired t-test with one tail. The required sample size of 29 participants was determined using G*Power software. However, recruiting 29 participants with low vision was not possible in this study, so the study was conducted with 16 participants in both groups.

Low vision is defined as visual acuity worse than 6/18 and visual fields less than 20° in diameter in the better-seeing eye with the best possible correction [30]. The meaning of 6/18 is that someone with normal vision can see letters at 18 m, but a person with low vision can see letters at 6 m [31]. Based on visual acuity according to the definition of low vision, participants were divided into two groups: Group A (n = 16, visual acuity less than 6/18) and Group B (n = 16, visual acuity greater than 6/18).

#### 2.2.2. VCDA Test App

For the visual information recognition speed test, we selected the top 20 downloaded apps (e.g., find a pharmacy and blood donation information) published on the Korean public data portal, and extracted 20 test screen samples. The 20 test screens produced 20 screens before applying VCDA (20 views, 20 questions, Q1–Q20) and 20 screens after applying VCDA (20 views, 20 questions, Q1–Q20). Then, we developed a test app that showed 20 screens in succession (Figure 1).

#### 2.2.3. Procedure for Data Collection

We showed 20 different mobile screens to the 32 participants and checked the visual information recognition speed using the stopwatch timer of the clock app installed in mobile phones. Each participant saw 10 mobile screens which had not adopted VCDA and 10 VCDA adopted mobile screens. The time taken for information recognition for each mobile screen was measured by two people to reduce the error. In addition, to achieve the accuracy of time measurement, each participant’s voice was recorded, and cross-validation was performed (Figure 2).

A different screen was shown each time to prevent an experimental effect of familiarity.

#### 2.2.4. Tools Used in This Study

##### Tools to Test Information Recognition Speed

To measure the time taken in the visual information recognition speed, we created 20 questions that could be answered only by accurately recognizing the information on the screen. Questions were placed at the bottom of each screen, and the measurement result was determined by the time the participant spent looking at the questions at the bottom and answering them (Table 3).

##### Satisfaction Survey

After the experiment, a satisfaction survey on VCDA was conducted with the participants. It consisted of a total of 9 questions and was received anonymously from the 32 participants. The questionnaire items of the satisfaction survey included assessing satisfaction with the VCDA design, satisfaction with the VCDA function, whether additional research on VCDA will be conducted in the future, and other aspects.

#### 2.2.5. Statistical Analysis

Analyses were performed using the SPSS for Statistical Computing (IBM SPSS 25.0 version) package, and Microsoft Excel was used for data collection. The Mann–Whitney test was used to compare the speed of the two groups, and the Wilcoxon signed-rank test was used to compare the before and after scores in the within group. Statistical significance was established if the *p*-value < 0.05.

## 3. Results

### 3.1. Participant Demographics

Participants in group A were recruited on a visual acuity basis, so the recruitment results were disproportionate. However, recruitment for participants in Group B did not prioritize visual acuity; instead, participants with as many diverse characteristics as possible were recruited (Table 4).

### 3.2. Comparison of Visual Information Recognition Speed between Group

Statistical analysis was conducted on the measurement results of Groups A and B (Table 5). As a result, there was no statistically significant difference (*p* > 0.05), but when the average value of the measurement time was compared, a practically significant difference of about 30 s was shown.

Moreover, when VCDA was applied, the difference in time before and after was statistically analyzed. The results showed that Group A experienced an almost statistically significant difference (*p* = 0.05), but Group B did not (*p* > 0.05).

### 3.3. Improvement Level of Visual Recognition Speed after VCDA by Items

Looking at the results for each question, when Group A used VCDA, three cases showed a statistically significant decrease (Q4, Q8, Q20), 15 showed a decrease (Q1, Q3, Q5, Q7, Q9-Q15, Q18, Q19), and two were maintained (Q16, Q17). In group B, one case showed a statistically significant decrease (Q10), 18 showed decreases (Q1–Q9, Q11–Q17, Q19, Q20), and one was unchanged (Q18) (Table 6, Figure 3).

Figure 4 shows how close the visual information recognition speed of people with normal vision was to people with low vision using VCDA.

### 3.4. Satisfaction Survey Results

After the experiment, an anonymous survey on satisfaction with VCDA was administered to the participants (Table 7). The results showed that with 4.08 points out of 5, most participants positively evaluated their visual information recognition due to VCDA.

However, the question regarding VCDA design satisfaction resulted in a slightly lower score than other items. Additionally, a slightly lower score was obtained for introducing the option to select the VCDA application (Figure 5).

## 4. Discussion

This study aimed to improve the visual information recognition speed of people with low vision. For this, we developed VCDA apps for the public open government data of Korea to examine the effect of information recognition speed on people with low vision. Visual information recognition speed was measured by analyzing the time taken before and after the application of VCDA in a sample of 32 participants, 16 with low and 16 with normal vision, to verify its effectiveness.

First, the time taken for visual information recognition before applying VCDA was 160.76 s for participants with low vision and 130.92 s for participants with normal vision. That is, the difference was 29.84 s. The difference between the two may seem minor but is a substantial gap. This outcome indicates that a person with low vision consistently loses 29.84 s whenever they engage in activities. This gap in visual information recognition speed will result in various social problems.

Next, when the participants with low vision used VCDA, the visual information recognition time was reduced by 20.12 s, from 160.76 to 140.64. Due to the small sample size, no statistically significant difference (*p* = 0.05) was detected. However, the study showed that visual recognition speed improvement approached significance for people with low vision after adopting VCDA. Interestingly, when using the VCDA developed for people with low vision, those with normal vision also showed an improved speed of information recognition. When a person with normal vision looked at a screen to which VCDA was applied, only the result for Q18 remained unchanged. The visual information recognition speed of all other items decreased. Thus, our results are meaningful and suggest it is possible to improve the information recognition speed for people with both low and normal vision.

Finally, regarding the satisfaction survey results, most respondents answered that VCDA helped in information recognition. Furthermore, they responded affirmatively concerning whether follow-up research should be conducted in the future. However, for the question of whether the existing screen is better than the version for people with low vision and if the VCDA was aesthetically pleasing, the score was slightly lower than for other items. Thus, the design of the VCDA was less aesthetically satisfying. On listening to the opinions of the participants on reasons for these complex responses, the primary cause was determined to be a reflection of the user’s experience. A user accustomed to an existing screen may find an unfamiliar one less satisfactory. Additionally, it is possible that the complex action required to use it led to this response. Based on the survey results, a change in user perceptions may be needed to improve the aesthetic satisfaction of GUI design for people with low vision.

This study has two limitations. First, it is challenging to recruit research participants with low vision. An adequate sample size for statistical analysis was not achieved. Second, the study recruited participants from only one region of Korea. Thus, this study’s results may reflect that region’s characteristics and may not be generalizable. To address these two limitations, a follow-up study must be performed by recruiting more research participants from various regions to replicate this experiment.

In conclusion, this study contributes significantly to research on addressing the mobile screen needs of those with low vision. It lays the foundation for research that can help reduce the gap in visual information recognition speed between people with low and normal vision through improving GUI software without the help of hardware aids.

## 5. Conclusions

This study developed VCDA guidelines for people with low vision as a method to enhance visual information recognition speed for this population. Experiments were conducted with people with low and normal vision using VCDA, and the guidelines were applied. As a result, speed improvement was observed for both low vision and normal vision groups that used VCDA. Therefore, the information recognition speed of people with low and normal vision can be improved by modifying the software (GUI) without changing the hardware (e.g., magnifying device).

## Figures and Tables

**Figure 1 healthcare-11-01047-f001:**
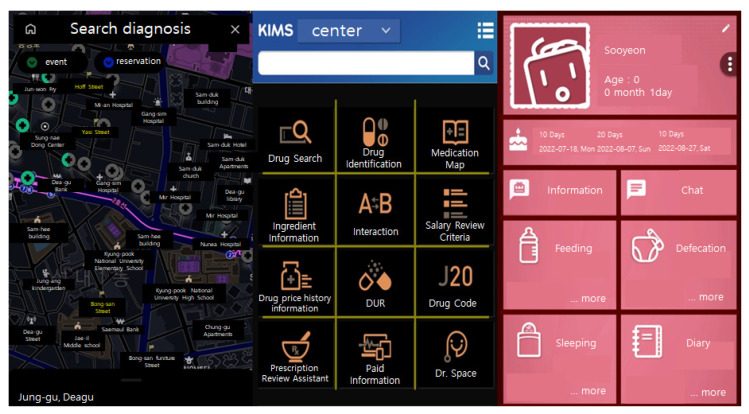
Sample of VCDA test app screenshot.

**Figure 2 healthcare-11-01047-f002:**
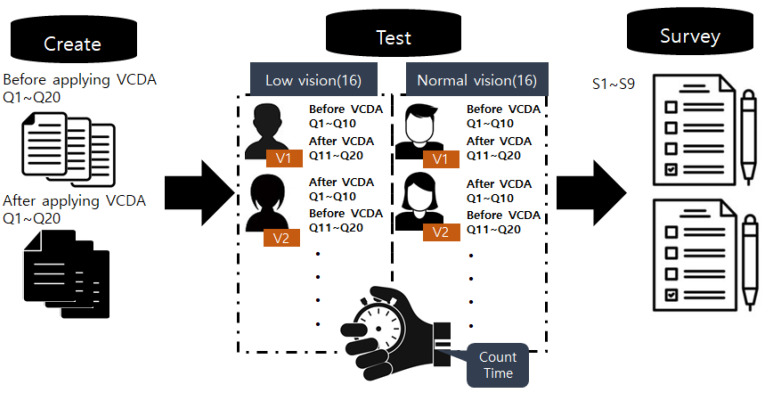
Information recognition speed test procedure.

**Figure 3 healthcare-11-01047-f003:**
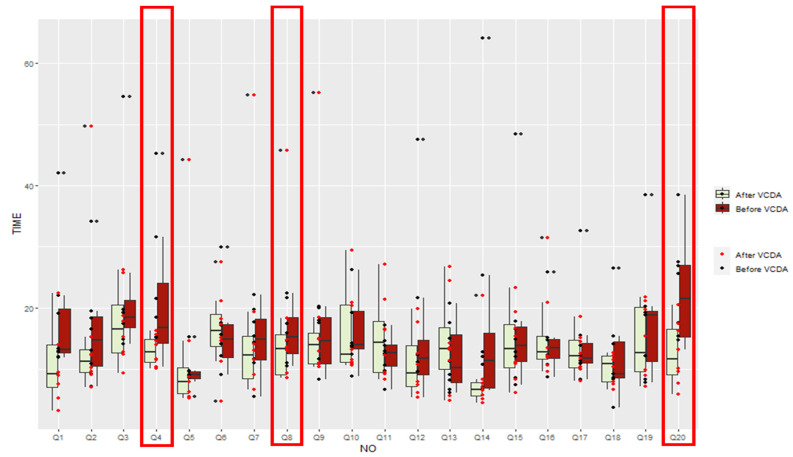
Comparison of information recognition speed before and after VCDA in Group A (low vision).

**Figure 4 healthcare-11-01047-f004:**
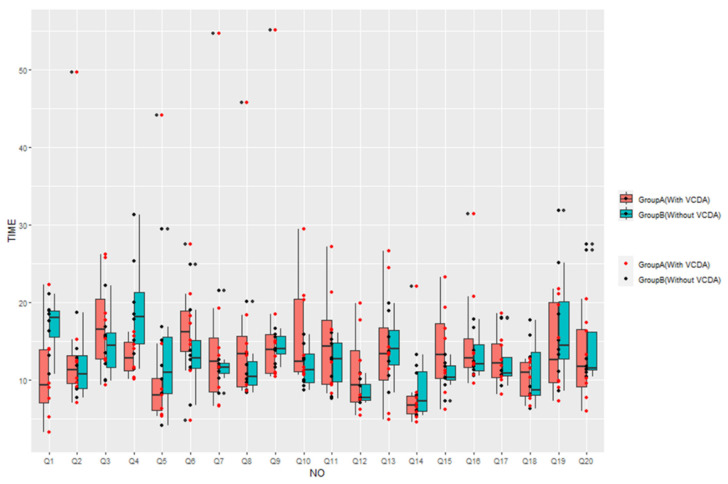
Comparison of information recognition speed between Group A (with VCDA) and Group B (without VCDA).

**Figure 5 healthcare-11-01047-f005:**
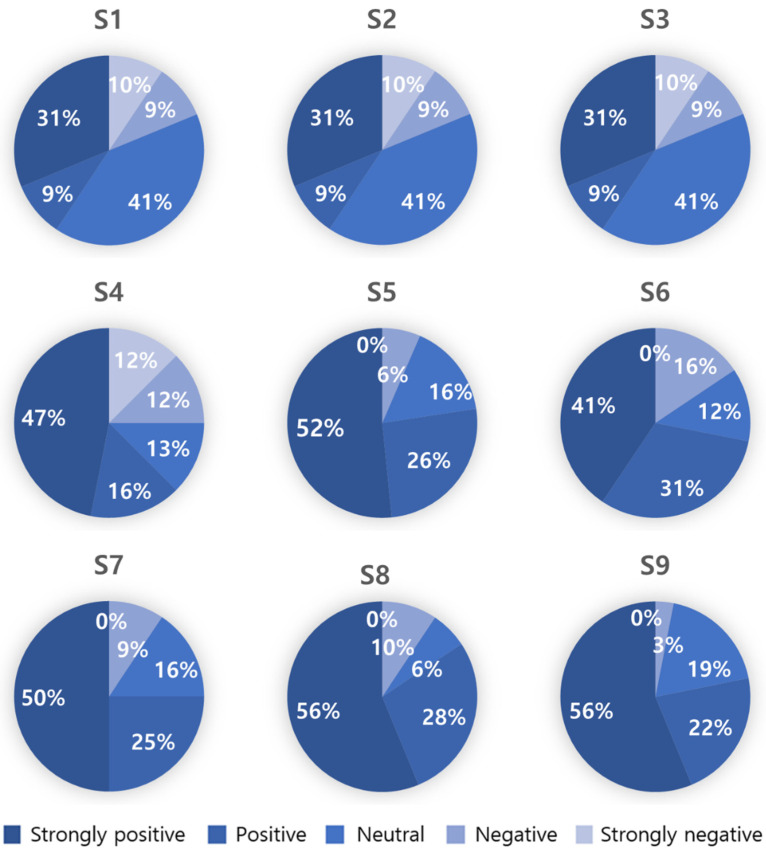
Satisfaction survey results with VCDA.

**Table 1 healthcare-11-01047-t001:** UXDG (user experience design guidelines) for the visually impaired [24].

**Increased data exposure**	Support for zoom in/out for the main content.
	Support for alternative color schemes.
**Data accessibility**	Intuitive navigation.
	The menu intuitively responds to user intentions.
	Support for input methods other than touch.
**Information acquisition**	Use of highly legible fonts.
	Highlighting main images that users can access.
	Highlighting action-triggering media.
**Data search**	Always placing text input boxes in the same location.
	Showing search results immediately for the search text box.
	Speech recognition for text input.

**Table 2 healthcare-11-01047-t002:** Visual Communication Design Accessibility (VCDA) Guidelines for Low Vision.

**Readability of text**	(a) Text size and thickness support more than three options.
	(b) Designated background colors that are contrasted with white characters.
	(c) Batch layout considering text-to-speech (TTS).
**Accessibility to Media (Image/Video)**	(d) Output alternate text or provide a TTS-listening button.
	(e) The characters in the image are recognized by optical character recognition (OCR) systems.
	(f) Provide a text-only button if the body text of information, such as media, is supported.
	(g) If a movie is included, notify its playability as alternative text.
**Simple, intuitive** **navigation**	(h) State the current page position in a consistent position for the entire information.
	(i) Always provide the same menu (e.g., Home or Top) to return to the first place.
	(j) The input area of all buttons is defined as 1.5 times the size of the button.
	(k) Provide a text-oriented black-and-white screen based on user experience.

**Table 3 healthcare-11-01047-t003:** Questions in the VCDA test app.

Q1	List the names of the two menus to the right of “First Aid Tips.”
Q2	List the names of the two pharmacies at 507 m and 640 m.
Q3	Name the pharmacy in Gaenari Park and in Uzukatsu Restaurant.
Q4	List the three departments on the left side of “Plastic Surgery” right below “Hospitals You Frequently Visit.”
Q5	Among the menus at the top, name the menu to the right of “Find Pharmacy.”
Q6	There are three buttons in the popup window. Please explain the location of the “Resend Verification Code” button among them.
Q7	Please tell us the names of all three menus under “Present Baby” menu when checking insurance premiums.
Q8	What input is being asked for under the picture of the baby? (Read the text)
Q9	How many times has Mr. Lim donated blood? Tell us the date when the next blood donation is possible.
Q10	As of 9 July 2022, please tell us all the remaining blood types on “10.0 days” and “10.7 days” of the blood holding status.
Q11	Please tell me the names of both menus on the “Prognosis Management” menu, arranged side by side.
Q12	This page is a screen introducing the service of the app. What is the name of this app?
Q13	Please tell me how many subway lines pass through Gyeongbuk Elementary School and Kangsim Hospital.
Q14	How many points does Mr. Lim have?
Q15	Please tell me the schedule for the two days after Constitution Day.
Q16	Please tell me the names of the three menus on the top row of the “Encyclopedia of Dementia.”
Q17	Please tell me the four menus on the far right (vertical bar) in order.
Q18	Is it necessary to agree to the “optional agreement” in the Terms of Use to sign up for membership?
Q19	Are you currently taking any medications? What should you press on the screen to add medications?
Q20	On the app screen, how many days ago was “Sooyeon” born? Also, does the app have a menu where you can write a diary entry?

**Table 4 healthcare-11-01047-t004:** Characteristics of the participants in each group.

Characteristics	Group A *	Group B *
Gender		
Male	7	6
Female	9	10
Age(year)		
20–29	11	3
30–39	4	7
40–49		3
50–59		1
60–69	1	2
Vision		
<6/18	16	
>6/18		16

* Group A: Low vision, Group B: Normal vision.

**Table 5 healthcare-11-01047-t005:** Statistical comparison of time taken for information recognition between and within group before and after VCDA.

	Before	After	Z ^1^	*p*-Value
Group A *	160.76	140.64	−1.96	0.05
Group B *	130.92	129.21	−0.70	0.48
Z ^2^	−1.47	−0.42		
*p*-value	0.14	0.67		

* Group A: Low vision, Group B: Normal vision. ^1^ Wilcoxon signed-rank test. ^2^ Mann–Whitney test.

**Table 6 healthcare-11-01047-t006:** Comparison of decrease in time for each question after applying VCDA.

	Significantly Improved ^1^	Practically Improved	Maintained
Group A *	Q4, Q8, Q20 (3)	Q1, Q2, Q3, Q5, Q7, Q9, Q10, Q11, Q12, Q13, Q14, Q15, Q18, Q19 (15)	Q16, Q17 (2)
Group B *	Q10 (1)	Q1, Q2, Q3, Q4, Q5, Q6, Q7, Q8, Q9, Q11, Q12, Q13, Q14, Q15, Q16, Q17, Q19, Q20 (18)	Q18 (1)

* Group A: Low vision, Group B: Normal vision. ^1^ *p*-value < 0.05.

**Table 7 healthcare-11-01047-t007:** VCDA satisfaction survey results.

Question	Mean	Standard Deviation
[S1] Do you think the VCDA proposed in this study is functionally valuable?	4.25	1.05
[S2] Do you think the VCDA helped you recognize information compared to a normal screen?	3.97	1.00
[S3] From an aesthetic perspective (personal design taste), do you think a normal screen is more aesthetically satisfying than the VCDA?	3.44	1.29
[S4] From this point of view, do you think developing a VCDA to “improve information awareness” is worthwhile compared to a normal screen designed with the “aesthetic satisfaction” of the information?	3.72	1.49
[S5] Do you think applying the VCDA on all public data apps by default would improve visual information access for low vision?	4.23	0.96
[S6] Do you think it is reasonable to mandate the provision of the VCDA option to provide accessibility (e.g., in-app settings) in current public data apps with the normal screen developed for normal-vision people?	3.97	1.09
[S7] Do you think that investment in the study should be made if the VCDA does not affect information access for normal-vision people but only helps people with low vision?	4.16	1.02
[S8] Do you think the research that decreases the information recognition speed of public government data should continue in the future?	4.31	0.97
[S9] Do you think further research is needed to increase accessibility for all social communities, not just for those with low vision?	4.31	0.90
Total	4.08	0.81

## Data Availability

The data presented in this study are available on request from the corresponding author. The data are not publicly available due to privacy.

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
