# Peer review of "Effects of Visual Communication Design Accessibility (VCDA) Guidelines for Low Vision on Public and Open Government Health Data"

_healthcare, 2023, doi:10.3390/healthcare11071047_

Round 1

Reviewer 1 Report

Authors presents an interesting manuscript entitle “Visual Communication Design Accessibility Guidelines for Improving Visual Information Recognition Speed of Public and Open Government Health Data for People with Low Vision” to “propose an improved model 18 of public data visual information accessibility guidelines for people with low vision through the 19 development of a graphic user interface (GUI) viewer for low vision (GVL) test app”. This is an interesting topic with potential application in other countries. 

This manuscript presents a comparison of low vision patients and normal vision subjects of vision performance (speed, etc.) that has potential to be published in Healthcare journal after a review. 

Current manuscript presentation must be improved previously to be accepted in Healthcare journal. So, Authors must improve some of the issues highlighted to facilitate future readers a better understanding of how this manuscript was conducted and the impact of its results.

The strengths of the article: Proposal of graphic user interface (GUI) viewer for low vision to facilitate access of information to these group of patients.

The weaknesses of the article: Study presentation must be clarified, manuscript could be summarized, methods need be better described (and include a statistical analysis of the results), results must include statistical comparison (P value) and be improved with dispersion value results (standard deviation, 95% confidence interval, interquartile range, etc.). Finally, conclusions must be deeply improved to be focused in study results.

This reviewer apologizes if he missed any important points or misinterpreted the goals of the study.

1) General comments

First issue of this report is related with use of terms low vision and vision disability. Because both are not synonymous. It is clear that low vision patients have a severe vision disability, but there is another vision disabilities (reduction of visual acuity and or visual field, colour vision, glare, binocular disorders, etc.) that have impact in vision performance but is not low vision.

So, Authors must decide if manuscript is focus in low vision patients or in persons with visual impaired (see Line 90 headlines 2.2, or Tables 1 & 2 headlines for example). Also, this Reviewer recommends to include low vision definition (described in Line 100) at Line 91 (at the beginning of 2.2 section). Moreover low vision definition must be improved. See definition provided by Tong J, Huang J, Khou V, Martin J, Kalloniatis M, Ly A. Topical Review: Assessment of Binocular Sensory Processes in Low Vision. Optom Vis Sci. 2021 Apr 1;98(4):310-325. doi: 10.1097/OPX.0000000000001672 “Low vision is defined as visual acuity worse than 6/18 and visual fields less than 20° in diameter in the better-seeing eye with best possible correction”.

In this paragraph some examples of conditions that provoke low vision (amblyopia, eye diseases, etc.) could be welcome. 

This issue is clear in Lines 115 to 129 focused in visually impaired people that could be different of low vision patients. 

Second issue is related with methods description. 

Can Authors provide a little more information about participants, for example age (mean, standard deviation and range, sex distribution, etc.) to demonstrate that both groups are comparable? if groups showed differences, this must be discussed as study limitation. Also, visual acuity of low vision patients is necessary (to clear presentation of participants).

Survey used in this study must be presented in Method section (number of items, questions, etc.). But is just cited (in Figure 2 and R1 description).

Methods and Results presentation should follow same order of presentation. Lines 289 to 300 summarize 3 main results, R1 is described in 3.1 section, but R1 describes the use of survey that is discussed in section 3.3. Please revise R3 description to match with 3.3 section.

Line 306 describes statistical analysis, but it is clearly inadequate. Statistical description must include a clear description of the software used (R) but also version, if normality of data were assessed and how, descriptive description of data (mean, median, standard deviation, interquartile range, percentage, etc.) and what type of statistical test was used to compare the differences between groups. All this information is compulsory in any research paper and also must match with results presentation.

Third issue is related with result presentation.

Please provide descriptive results of main data assessed or collected in this study (mean, median, standard deviation, 95% confidence interval, interquartile range, percentage, etc.). Statistical comparison of differences between groups of main assessed variables must be provided with the P value to support if exits (or not) statistical differences between groups. All figures must include error bars of 95% confidence interval or interquartile range to clear results presentation. Also, in figures it is highly recommended include or highlight if differences are statistically significant (for example with a symbol or with the P value).

Figures must be clear for readers and figure legend must clearly describe results. For example Figures 5 and 6 showed a lot of “points” that could be “outlier” but this must be described in figure legend (because some readers can understand that asterisks mean statistically significant differences, for example and be confused). Also, this Reviewer encourages to highlight is any statistical difference is detected in Figures 5 and 6, because if not differences exit the utility of both figures could be limited. Also, what is exactly the mean of Q1-20, survey has 10 questions so this presentation must be clarified. Should be that Q1-20 represents GUI for low vision? This Reviewer recommends use different description of GUI that used for questions of the survey, for example (GUI-1, GUI-2 could help to future readers to understand information in each part of the manuscript. If Authors change description of GUI in Figures 5 and 6 this change must be placed in Material and method section as well (Lines 268-275).

In this line, Figures 7, 8 & 9 are also unclear. Is information of both groups represented? Why not answer of each group are summarized and compared -for example with a chi square test-? This Reviewer recommends revise these three figures and clarify what results are summarized.

Fourth issue is related with conclusions.

Lines 478-500 are not study conclusions; these paragraphs present a description of current situation (that is Introduction) and some Authors opinion (that is Discussion). Please, rewrite conclusion with information supported with study results, differences in speed between groups with GUI and if this study proposal could be applied in other context or countries.

Minor changes

Please provide a reference to support information in Table 1 and clarifies if this table is regarding low vision of is related with visual impairments. For example, colour vision abnormalities not always are concomitant to low vision.

Line 172 replace poor by low.

Figure 3 must be deleted because it is easy to summarize in a simple sentence.

Deviation data (standard deviation, etc.) and P value must be provided in all results presented, whe is appropiated. 

Lines 320-325. These paragraphs are not study results must be deleted of moved to other manuscript section if is applicable (may be Discussion).

Author Response

Dear Reviewer 1:

I wish to re-submit the manuscript titled “Visual Communication Design Accessibility Guidelines for Improving Visual Information Recognition Speed of Public and Open Government Health Data for People with Low Vision.” The manuscript ID is healthcare-2181109.

We thank you and the reviewers for your thoughtful suggestions and insights. The manuscript has benefited from these insightful suggestions. I look forward to working with you and the reviewers to move this manuscript closer to publication in the Journal Name in Italics.

The manuscript has been rechecked and the necessary changes have been made in accordance with the reviewers’ suggestions. The responses to all comments have been prepared and are attached herewith.

Thank you for your consideration. I look forward to hearing from you.

Sincerely,

[Author’s name] Mr. Jongho Lim

[Affiliation] Kyungpook National University

[Postal address] 41566

[Phone number] +82-10-9259-2852

[Email address] [email protected]

Reviewer 2 Report

The manuscript presents a set of guidelines for improving the visual accessibility of mobile apps. The relevance and importance of the work are clear, as it becomes evident that people with low vision need more time to interpret visual cues. The methods are well-described, with some few mistakes, as follows:

- throughout the manuscript, the percent symbol shall be separated from the value (e.g. "337 %" instead of "337%");

- it makes no sense to calculate percentual increases or reductions over percentual values. The authors should use "pp - percentual points", and subtract the values (e.g., from 18.56 % to 8.12 % there is a reduction of 10.44 pp). See also lines 354-356;

- there should be a paragraph at the end of the Introduction depicting the paper structure;

- in Table 2, the item "The characters in the image are optical character recognition (OCR)." makes no sense. Shouldn't it be something like "The characters in the image are recognized by optical character recognition (OCR) systems"? Same in line 185;

- line 165 makes no sense ("users can choose this can be used as a guideline to provide a consistent layout for...");

- the sentence "...size, which is a clear screen intended by the planner, developer, and designer", in lines 168/169, makes no sense;

- in line 177, shouldn't the flow be from top left to bottom right?

- in line 191, wouldn't the term "text-only" be more clear than "read-only"?

- in line 201, the sentence "you may not know if you need to press it or if you can click it." is confusing. What is the difference between "press it" and "click it"?

- in line 220, there is repeated text ("effects such as invisible text and invisible text. ")

- in Table 5, the terms "unapplied originals screens" and "applied screens" seem odd;

- in Table 5, first line of the Method, shouldn't it be Q1 to Q20?

- there is a typo in figure 1 ("normal viewer");

- in line 286, Table 4 is wrongly referred to (It should be Table 5);

- in the figures with time values, the symbol for the "second" unit is "s", not "sec";

- in line 314, the sentence "It was found that the two had an information recognition speed of approximately 29.84 seconds." is wrong. This is the DIFFERENCE in the TIME needed to recognize the text;

- in section 3.2, first the raw data should be presented (Figure 4), and THEN the conclusions should be drawn;

- in figures 7-9, the colors are misleading, and should have a clear intensity scale;

Aside from the above minor issues, the paper has some serious shortcomings in the analysis of the results, which prevent its publication, as follows:

- throughout the Results section, the authors confound the quantities "speed" and "time". For instance, Figure 3 does not show the "speed gap", but the "time difference";

- I disagree with the calculation of 18.56 % difference. If the reference is Group B time (130.92 s), then Group A time (160.76 s) is 22.79 % higher. It seems the authors calculated the other way around, but then they should state that Group B has an average time 18.56 % LOWER than Group A.

- I disagree with the conclusions of paragraph 321-323. This calculation assumes that all people are constantly, over 70 years, recognizing visual information. The authors should consider an estimate of the percent of daily time used for such tasks, and then make the comparison. Even if the assumption was correct, the value should be 70 x 160.76 / 130.92 = 85,96 years.

- The main shortcoming of the manuscript is the lack of a proper statistical analysis of the data. The authors only compared the average values, but this is a clear case of applying hypothesis t-tests (paired or not, depending on the case) and estimating p-values, which would confirm (or not) the hypothesis of the authors.

Author Response

Dear Reviewer 2:

I wish to re-submit the manuscript titled “Visual Communication Design Accessibility Guidelines for Improving Visual Information Recognition Speed of Public and Open Government Health Data for People with Low Vision.” The manuscript ID is healthcare-2181109.

We thank you and the reviewers for your thoughtful suggestions and insights. The manuscript has benefited from these insightful suggestions. I look forward to working with you and the reviewers to move this manuscript closer to publication in the Journal Name in Italics.

The manuscript has been rechecked and the necessary changes have been made in accordance with the reviewers’ suggestions. The responses to all comments have been prepared and are attached herewith.

Thank you for your consideration. I look forward to hearing from you.

Sincerely,

[Author’s name] Mr. Jongho Lim

[Affiliation] Kyungpook National University

[Postal address] 41566

[Phone number] +82-10-9259-2852

[Email address] [email protected]

Round 2

Reviewer 1 Report

Authors presents the second version of the manuscript entitle “Visual Communication Design Accessibility Guidelines for Improving Visual Information Recognition Speed of Public and Open Government Health Data for People with Low Vision”. Authors have made a significant modification of their manuscript that now could be suitable to be published in Healthcare journal.

Authors submitted the document with control-track changes on. This presentation is useful to detect where changes have made if manuscript shows relatively small number of changes, but in this case a really large modifications have been done and reading manuscript with track-changes on it is very difficult and a “clean version of the manuscript” could be welcome to facilitate the assessment of this second version. It is important because Reviewers have access to a PDF (not word file). So, this Reviewer recommends a careful revision of manuscript proof to avoid possible mistakes in some sentences.

Also, exact P value in text (abstract and main manuscript text) could help to be clearest as possible, and use P>0.05 or P<0.05 just to summarize more than one P value.

This reviewer apologizes if he missed any important points or misinterpreted the goals of the study.

Author Response

(The authors gave the same response as above.)

Reviewer 2 Report

This is the second version of the manuscript, but in fact it could be considered as a new manuscript, as it has been so much changed (and reduced) that it not really a simple revision. In some aspects it has improved (especially the statistical analysis), but the authors opted to remove large sections of the manuscript instead of improving them, so it now lacks a proper contextualization. 

For instance, the entire section 2.4 of the original version was removed, and it is quite important content, as it details the previous works. It should be reinstated, preferably in the Introduction. In this sense, the paragraph 298-305 should actually be part of this introduction, and not be in the Discussion.

Figures 5 and 6 of the original version are also important to keep, as well as Table 6 and Figures 7-9 (just using a better color scheme). 

The rest of the removals and simplifications are not relevant and could be kept.

Finally, I did not understand the meaning of the sentence "If the subject sees a problem with the same information, the speed can be increased." in line 194.

Author Response

Dear Reviewer 2:
a

I wish to re-submit the manuscript titled “Visual Communication Design Accessibility Guidelines for Improving Visual Information Recognition Speed of Public and Open Government Health Data for People with Low Vision.” The manuscript ID is healthcare-2181109.

We thank you and the reviewers for your thoughtful suggestions and insights. The manuscript has benefited from these insightful suggestions. I look forward to working with you and the reviewers to move this manuscript closer to publication in the Journal Name in Italics.

The manuscript has been rechecked and the necessary changes have been made in accordance with the reviewers’ suggestions. The responses to all comments have been prepared and are attached herewith.

Thank you for your consideration. I look forward to hearing from you.

Sincerely,

[Author’s name] Mr. Jongho Lim

[Affiliation] Kyungpook National University

[Postal address] 41566

[Phone number] +82-10-9259-2852

[Email address] [email protected]
